# Research on Filtering Algorithm of MEMS Gyroscope Based on Information Fusion

**DOI:** 10.3390/s19163552

**Published:** 2019-08-15

**Authors:** Hui Guo, Huajie Hong

**Affiliations:** College of Intelligence Science and Technology, National University of Defense Technology, Changsha 410073, China

**Keywords:** MEMS gyroscope, line accelerometer, noise, drift, Kalman filter

## Abstract

As an important inertial sensor, the gyroscope is mainly used to measure angular velocity in inertial space. However, due to the influence of semiconductor thermal noise and electromagnetic interference, the output of the gyroscope has a certain random noise and drift, which affects the accuracy of the detected angular velocity signal, thus interfering with the accuracy of the stability of the whole system. In order to reduce the noise and compensate for the drift of the MEMS (Micro Electromechanical System) gyroscope during usage, this paper proposes a Kalman filtering method based on information fusion, which uses the MEMS gyroscope and line accelerometer signals to implement the filtering function under the Kalman algorithm. The experimental results show that compared with the commonly used filtering methods, this method allows significant reduction of the noise of the gyroscope signal and accurate estimation of the drift of the gyroscope signal, and thus improves the control performance of the system and the stability accuracy.

## 1. Introduction

A photoelectric stability platform is a device that can effectively isolate the carrier disturbance and keep the visual axis stable. As an important inertial sensor of photoelectric stability platform, the gyroscope is mainly used for the angular velocity of the sensitive inertia space of each axis. The measured angular velocity is fed back to the servo control system to form a speed closed loop, thereby isolating the external disturbance and ensuring the stability of the platform. The MEMS gyroscope is widely used in photoelectric stabilization systems due to its small size and low cost. However, due to the influence of devices and environmental interference, the output of the MEMS gyroscope has a certain noise and drift, which reduces the accuracy of the signal and affects the stability of the system.

At present, there are many methods for processing gyroscope signal noise, which can be roughly divided into two categories: one is the based on the model compensation method, that is, modeling the random noise of the gyroscope and offsetting it in accordance with the model, e.g., the Kalman filter method [1,2,3], statistical filtering, etc.; the other is the direct filtering method, which directly filters the output signal of the gyroscope, e.g., low-pass filtering, Wavelet filtering [4,5,6,7], adaptive filtering [8], and so on.

In the literature [9], based on the time series model of gyroscope zero drift data, the author uses the Kalman algorithm to process the drift data. The simulation results show that the method can effectively filter the static gyroscope signal, but the dynamic signal filtering effect is poor. Another paper [10] proposes an analytical method to estimate the measurement noise variance in Kalman filter and apply it to the Kalman filter in real time; experiments show that this method could carry out a more accurate result in estimating the measurement noise variance and improve the accuracy of Kalman filter; however, it is not able to estimate the drift of the sensor. In the literature [11], Luming Li combines Kalman filtering with neural networks to estimate the variance of the Kalman filter measurement noise through neural networks, thus solving the problem of inaccurate selection of measurement noise variance. Experiments show that the algorithm better suppresses the random noise of the gyroscope. However, since this method is also based on time series modeling, the filtering effect on dynamic signals is unsatisfactory. In the literature [12], the Kalman filter is used to fuse the two sensor signals of the MEMS gyroscope and the accelerometer to suppress the gyroscope random noise, meanwhile estimating the drift of the accelerometer. However, because the existence of the constant drift of the gyroscope is not taken into consideration in this algorithm, there is a certain error with the actual situation. In the literature [8], in order to eliminate the random noise of the fiber optic gyroscope, the digital low-pass filtering, wavelet filtering, adaptive filtering, and variable-step adaptive filtering methods are compared for filtering. The comparison shows that the variable step size adaptive filtering method has the best filtering effect on the gyroscope random noise. However, this method cannot simultaneously estimate the constant drift of the sensor.

For the estimation of gyroscope signals drift, the literature [13] uses six accelerometers placed symmetrically on the UAV’s rotary axis to calculate the angular velocity. The Kalman filter algorithm is used to fuse the angular velocity of the gyroscope output with the angular velocity calculated by the accelerometer to estimate the constant drift of the gyroscope. The simulation results show that the method can effectively compensate the constant drift of the gyroscope and has certain application value. However, this method has only a certain effect in the simulation and has not been verified in the experiment.

Although the above researches have achieved certain results on the filtering and drift estimation of the gyroscope signal, the filtering and drift estimation of the gyroscope noise often cannot be obtained at the same time. Although some scholars also use gyroscopes and line accelerometers to achieve gyro filtering and drift estimation, they ignore the existence of linear accelerometer drift, making the results less accurate. Based on the summary of the current research results, this paper uses the Kalman filter algorithm that combines the gyroscope signal and the accelerometer signal to process the gyroscope signal and also estimate the accelerometer signal filtering and drift, thereby improving the detection accuracy of the sensor.

## 2. Analysis of MEMS Gyroscope Error

### 2.1. Analysis of Noise Sources of MEMS Gyroscopes

The noise of MEMS gyroscopes is mainly caused by imperfect structure and environmental interference. In order to identify and characterize the noise source that causes the gyroscope noise, the Allan variance method is used to analyze the MEMS gyroscope signal. This method is characterized by the ability to characterize and identify the statistical properties of the entire noise, meanwhile analyzing the source of the identified noise term [8].

Let the length of the gyroscope signal be *L* and the sampling frequency be f, then the sampling period is $t_{0}=1/f$. The average gyroscope signal is divided into *K* groups (K≥2), then the data length of each group is N=L/K and the length of each group of data is $t=t_{0}*N$, which is also called the correlation time [14].

Calculate the average value $\bar{\omega_{k}}\left(N\right)$ of each group at each correlation time t, then
(1)$$\bar{\omega_{k}}\left(N\right)=\frac{1}{N}\sum_{i=1}^{N}\omega_{\left(k-1\right)*N+i}K=1,2,\cdots ,K$$

Then the Allan method at the relevant time t is defined as
(2)$$\sigma_{A}^{2}\left(t\right)=\frac{1}{2\left(K-1\right)}\sum_{j=1}^{K-1}{\left[\bar{\omega_{j+1}}\left(N\right)-\bar{\omega_{j}}\left(N\right)\right]}^{2}$$

The Allan variance curve can be obtained by taking different correlation times t and calculating the corresponding Allan variance $\sigma_{A}^{2}\left(t\right)$. The square root $\sigma_{A}\left(t\right)$ of the Allan variance is called the Allan standard deviation and the curve of $\sigma_{A}\left(t\right)\sim t$ in the double logarithmic coordinate system is called the Allan standard deviation double logarithmic curve [15].

The random drift error of MEMS gyroscope mainly includes five noise sources such as quantization noise, rate ramp noise, angular rate random walk noise, angular random walk noise, and zero offset instability noise [16]. Since these noise sources are reflected in different correlation time regions, the Allan variance method can analyze different noises of the gyroscope. The Allan variance of the quantization noise is $\sigma_{Q}^{2}\left(t\right)=3Q^{2}/t^{2}$. The Allan variance of the angular random walk noise is $\sigma_{N}^{2}\left(t\right)=N^{2}/t$. The Allan variance noise of the zero-bias instability is $\sigma_{B}^{2}\left(t\right)=2B^{2}ln2/\pi$. The Allan variance of the angular rate random walk noise is $\sigma_{K}^{2}\left(t\right)=K^{2}t/3$. The Allan variance of the rate ramp noise is $\sigma_{R}^{2}\left(t\right)=R^{2}t^{2}/2$. Among them, Q, *N*, *B*, *K*, and *R* are various error coefficients.

If the MEMS gyroscope contains these five noise sources and is statistically independent. Then the Allan variance can be decomposed into the sum of the Allan variances of the noise sources.
(3)$$\sigma_{A}^{2}\left(t\right)=\sigma_{Q}^{2}\left(t\right)+\sigma_{N}^{2}\left(t\right)+\sigma_{B}^{2}\left(t\right)+\sigma_{K}^{2}\left(t\right)+\sigma_{R}^{2}\left(t\right)$$
(4)$$\sigma_{A}^{2}\left(t\right)=\frac{3Q^{2}}{t^{2}}+\frac{N^{2}}{t}+\frac{2B^{2}ln2}{\pi }+\frac{K^{2}t}{3}+\frac{R^{2}t^{2}}{2}=\sum_{i=-2}^{2}\varphi_{i}t^{i}$$

Using the least squares method, $\varphi_{i}$ can be obtained and various error coefficients are obtained. The relationship between each error coefficient and $\varphi_{i}$ is
$$Q=\frac{\sqrt{\varphi_{-2}}}{3600\sqrt{3}},N=\frac{\sqrt{\varphi_{-1}}}{60},B=\frac{\sqrt{\varphi_{0}}\pi }{2ln2},K=60\sqrt{3\varphi_{1}},R=3600\sqrt{2\varphi_{2}}$$

In this paper, the static data of MEMS gyroscope is collected at the sampling frequency of 1 KHz. The analysis results are shown in Figure 1.

The error coefficients obtained by Allan analysis of variance are
$$Q=5.4412\times 10^{-7},N=8.0041\times 10^{-4},B=0.0258,K=3.973,R=121.3356$$

As can be seen from the above figure, the noise composition of the MEMS gyroscope is mainly composed of quantization noise and angular random walk noise. Therefore, the filtering of MEMS gyroscope noise is mainly to filter the two noise sources.

### 2.2. Influence of MEMS Gyroscope Error on Stable Platform System

The influence of the error of the MEMS gyroscope on the stable platform control system can be expressed by the mean square error of the system output under the influence of the noise and drift of the gyroscope. The system control model is shown in Figure 2.

In Figure 2, *C*(*s*) is the controller of the stable loop, *G*(*s*) is the transfer function of the system model, *H*(*s*) is the transfer function of the sensor, r(t) is the speed command entered by the system, v(t) is the speed of the system output, and n(t) is the error of the gyroscope.

According to the Figure 2, the transfer function of the gyroscope error input to the system output is
(5)$$G_{c}\left(s\right)=-\frac{H\left(s\right)C\left(s\right)G\left(s\right)}{1+H\left(s\right)C\left(s\right)G\left(s\right)}$$

From the linear system theory, the output power spectral density of a linear system is equal to the product of the input power spectral density and the system power transfer function [17]. According to the system shown in Figure 2, if the power spectral density of the gyro error is $\phi_{r}\left(\omega \right)$. The transfer function of the gyroscope noise input to the system output is $G_{c}\left(s\right)$. The power spectral density of the system output caused by the gyro noise input is
(6)$$\phi_{v}\left(\omega \right)={\left|G_{c}\left(j\omega \right)\right|}^{2}\phi_{r}\left(\omega \right)$$

According to Equation (6), the system mean square error caused by the gyroscope error is
(7)$$\varepsilon^{2}=\int_{-\infty }^{\infty }{\left|G_{c}\left(j\omega \right)\right|}^{2}\phi_{r}\left(\omega \right)d\omega$$

It can be seen from Equation (7) that the influence of the gyroscope error on the output of the system is determined by the power spectral density of the gyroscope error and its transfer function in the control system. When the transfer function of the control system is fixed, the larger the noise and drift of the gyroscope, the larger the mean square error of the system output due to it.

In addition, in the platform stability control loop, due to the presence of gyro noise, the controller parameters are often not further optimized, which limits the improvement of system control performance. The existence of constant drift of the gyroscope will cause the system to have a smaller amplitude control output even when the platform is stable, which causes the platform to rotate slowly, affecting its stability accuracy, and more serious may cause the system to not work properly.

Therefore, in order to reduce the influence of MEMS gyroscope error on the system and improve the control performance of the system, it is necessary to process the noise and constant drift of the gyroscope.

## 3. Design and Simulation of the Filter Algorithm

There are many filtering methods for gyroscopes, such as low-pass filter, Kalman filter, forward linear filter, and wavelet filter. Considering the feasibility and real-time requirements, the Kalman filter algorithm and forward linear prediction filter algorithm will be highlighted below. We compare the filtering effects of the two algorithms to find the best filter algorithm.

### 3.1. Improved Kalman Filter Algorithm

#### 3.1.1. Design of Kalman Filtering Algorithm Based on Information Fusion

Kalman filter is a filtering method based on the minimum mean square error criterion. It obtains an estimate of the state value through an iterative calculation. And it is the best estimate for linear systems with Gaussian white noise. Due to the high accuracy and good real-time performance of Kalman filter, it has been widely used in signal processing.

The equation of state for the discrete time process of the Kalman filter is expressed as
(8)$$x_{k}=Ax_{k-1}+Bu_{k-1}+F\omega_{k-1}$$

The measurement equation is expressed as
(9)$$y_{k}=Cx_{k}+Dv_{k}$$

In Equation (9), $x_{k}$ is the system state quantity at time k, $u_{k-1}$ is the system control quantity at time k − 1, ***A*** is the state transition matrix, ***B*** is the control matrix, ***C*** is the measurement matrix, and ***D*** and F are the noise matrix. $\omega_{k}$ is the system process noise, $v_{k}$ is the system measurement noise, and $\omega_{k}$ and $v_{k}$ are mutually independent, normally distributed white noise.
(10)$$\omega_{k}\sim N\left(0,Q\right)$$
(11)$$v_{k}\sim N\left(0,R\right)$$

***Q*** and R are the covariance matrices of system noise and measurement noise, respectively.

The steps of the Kalman filter algorithm are as follows.

State one-step prediction:
(12)$${\bar{x}}_{k|k-1}=A{\bar{x}}_{k-1}+Bu_{k-1}$$

One-step prediction of covariance matrix:
(13)$$P_{k|k-1}=AP_{k-1}A^{T}+Q$$

Calculate the filter gain:
(14)$$K=P_{k|k-1}C^{T}\left(CP_{k|k-1}C^{T}+R\right)$$

State estimation:
(15)$${\bar{x}}_{k}={\bar{x}}_{k|k-1}+K\left(y_{k}-C{\bar{x}}_{k|k-1}\right)$$

Covariance matrix estimation:
(16)$$P_{k}=\left(I-KC\right)P_{k|k-1}$$

According to the motion relationship, when the angular velocity value $v_{k}$ and the angular acceleration value $a_{k}$ of the system at time k are known, the angle and angular velocity of the system at time k + 1 can be obtained as follows
(17)$$x_{k+1}=x_{k}+v_{k}*t+\frac{1}{2}*a_{k}*t^{2}$$
(18)$$v_{k+1}=v_{k}+a_{k}*t$$

According to the Formulae (17) and (18), it is possible to construct a state equation for Kalman filtering using information such as angle, angular velocity, and angular acceleration. This avoids the use of ARMA models or state equations of the original system that are difficult to obtain for Kalman filtering.

Since both the gyroscope and the accelerometer have noise and drift, in order to improve the algorithm’s effect on sensor noise reduction and estimation of sensor drift, the state of the Kalman filter is augmented as $X={\left[\begin{array}{ccc} x & v & \begin{array}{ccc} \Delta v & a & \Delta a \end{array} \end{array}\right]}^{T}$. Where x is the angle information, v is the angular velocity information, Δv is the angular velocity drift of the sensor, a is the angular acceleration information, and Δa is the angular acceleration drift of the sensor. Therefore, the system equations and observation equations after the amplification state are
(19)$$\{\begin{array}{c} X_{k|k-1}=AX_{k-1}+BW_{k-1}\\ Y_{k}=CX_{k}+V_{k} \end{array}$$
where
$$A=\left[\begin{array}{ccccc} 1 & T & -T & \frac{T^{2}}{2} & -\frac{T^{2}}{2}\\ 0 & 1 & 0 & T & -T\\ 0 & 0 & 1 & 0 & 0\\ 0 & 0 & 0 & 1 & 0\\ 0 & 0 & 0 & 0 & 1 \end{array}\right],B=\left[\begin{array}{ccccc} 1 & 0 & 0 & 0 & 0\\ 0 & 1 & 0 & 0 & 0\\ 0 & 0 & 1 & 0 & 0\\ 0 & 0 & 0 & 1 & 0\\ 0 & 0 & 0 & 0 & 1 \end{array}\right],C=\left[\begin{array}{ccccc} 1 & 0 & 0 & 0 & 0\\ 0 & 1 & 0 & 0 & 0\\ 0 & 0 & 0 & 1 & 0 \end{array}\right]$$
where T is the sampling time.

The angle information can be solved by the gravity component to which the linear accelerometer is sensitive. The angular velocity information is measured by a gyroscope. The angular acceleration information is measured by a linear accelerometer. It can be shown in Figure 3.

In Figure 3, two linear accelerometers are placed symmetrically about the axis of rotation. *G* is the acceleration of gravity;
$a_{1}$ and $a_{2}$ are the output values of the two linear accelerometers, respectively; *d* is the distance of the line accelerometer from the axis of rotation; θ is the angle at which the platform rotates; and α is the angular acceleration of the platform rotation. According to the relationship, you can get
(20)$$a_{1}=\alpha d-Gcos\theta$$
(21)$$a_{2}=-\alpha d-Gcos\theta$$

The above formula is combined
(22)$$\alpha =\frac{a_{1}-a_{2}}{2}$$
(23)$$\theta =\arccos\left(-\frac{a_{1}+a_{2}}{2G}\right)$$

Thus the angle and angular acceleration of the platform can be obtained.

It can be deduced from the above that the improved Kalman filter algorithm can not only filter the gyroscope signal and estimate its constant value drift, but also filter the acceleration signal and estimate the constant value drift of the accelerometer. The processing of the acceleration signal is to better achieve the algorithm’s filtering of the gyroscope signal and the estimation of the gyroscope drift.

#### 3.1.2. Proof of Stability of Kalman Filter Algorithm Based on Information Fusion

The stability problem of the Kalman filter is related to its application in practical engineering. Therefore, the stability of the filter should be verified after the filter is designed. The filter stability problem requires studying the influence of the initial value of the parameters on the filter stability. That is, as the filtering time increases, the estimated value of the state and the error variance matrix are not affected by the selected initial value [18]. To prove the stability of the filter, the following theorem is used:

Filter stability theorem: If the system is fully controllable and fully observable, the Kalman filter is uniformly progressively stable [19,20].

Among them, the observability and controllability criteria of linear systems are as follows.

(1) Linear system controllability criteria: For linear constant discrete system
(24)x(k+1)=Ax(k)+Bu(k)

The necessary and sufficient condition for complete controllability is that the rank of the matrix $\left[\begin{array}{cccc} B & AB & \cdots & A^{n-1}B \end{array}\right]$ is n.

(2) Linear System Observability Criterion: For linear constant discrete system
(25)$$\{\begin{array}{c} x\left(k+1\right)=Ax\left(k\right)+Bu\left(k\right)\\ y\left(k\right)=Cx\left(k\right)+v\left(k\right) \end{array}$$

The necessary and sufficient condition for complete observability is that the rank of the matrix ${\left[\begin{array}{cccc} C & CA & \cdots & CA^{n-1} \end{array}\right]}^{T}$ is n. Where n is the dimension of the state vector.

According to Equation (17),
(26)$$\mathrm{rank}\left[\begin{array}{ccccc} B & AB & A^{2}B & A^{3}B & A^{4}B \end{array}\right]=5$$
(27)$$\mathrm{rank}{\left[\begin{array}{ccccc} C & CA & CA^{2} & CA^{3} & CA^{4} \end{array}\right]}^{T}=5$$

As can be seen from the above equation, the Kalman filtering algorithm proposed in this paper is completely controllable and fully observable. According to the filter stability theorem, the algorithm is stable.

### 3.2. Forward Linear Prediction Filter Algorithm

The principle of the forward linear predictive filter algorithm is to multiply the previous gyroscope signal by the corresponding weight to predict the gyroscope signal at the current time. The algorithm begins with an initial value and then calculates the difference between the current gyroscope actual value and the predicted value. The weight vector is adjusted according to the minimum mean square error algorithm to converge to a stable weight [21].
(28)$$\hat{x}\left(n\right)=\sum_{i=1}^{K}a_{i}x\left(n-i\right)=A^{T}X\left(n-1\right)$$

$X\left(n-1\right)={\left[x\left(n-1\right)\;x\left(n-2\right)\;\cdots \;x\left(n-K\right)\right]}^{T}$ is the vector of the gyro signals from the first *K* moments; $a_{i}$ is the weight and K is the order of the filter.

The difference between the current gyroscope actual value and the current gyroscope predicted value and the cost function are
(29)$$e\left(n\right)=x\left(n\right)-\hat{x}\left(n\right)$$
(30)$$J\left(n\right)=E\left(e^{2}\left(n\right)\right)$$

According to the theory of minimum mean square error, J(n) should take the minimum value if the best weight is to be obtained. Therefore, the weight adjustment equation is
(31)A(n+1)=A(n)+μE[e(n)X(n−1)]

In the actual calculation, in order to reduce the amount of calculation, the above equation is simplified as
(32)A(n+1)=A(n)+μe(n)X(n−1)

In the equation, μ is the step factor, which is generally a smaller value greater than zero [22].

### 3.3. Simulation Analysis of Filter Algorithm

For the filtering algorithm designed above, the closed-loop control system shown in Figure 4 is built for simulation analysis. Where *r(t)* is the angular velocity command signal, *G*(*s*) is the transfer function of the platform system, and *C*(*s*) is the controller of the system speed closed loop. The angular velocity noise signal *vn(t)* added by the system is a white noise signal with a variance of 1 ${\left(\deg/s\right)}^{2}$ and an average of 0.5 deg/s. The added acceleration noise signal *an(t)* is a white noise signal with a variance of 5 ${\left(\deg/s^{2}\right)}^{2}$ and an average value of 1 ${\left(\deg/s^{2}\right)}^{2}$.

#### 3.3.1. Analysis of Static Filtering

Set the angular velocity command to zero. The acquisition of the gyroscope raw signal, the Kalman filtered signal, and the forward linear filtered signal are shown in Figure 5, Figure 6, Figure 7, Figure 8 and Figure 9.

It can be seen from Figure 5, Figure 6, Figure 7, Figure 8 and Figure 9 that the Kalman filter algorithm and the forward linear filter algorithm have the effect of reducing the gyroscope noise level when the gyroscope noise level is constant. Among them, the Kalman filter algorithm reduces the variance of the noise from 1 ${\left(\deg/s\right)}^{2}$ to 0.0056 ${\left(\deg/s\right)}^{2}$. The forward linear filtering algorithm reduces the variance of the noise from 1 ${\left(\deg/s\right)}^{2}$ to 0.0119 ${\left(\deg/s\right)}^{2}$. It can be seen that the Kalman filter effect is better than the forward linear filter effect. Moreover, the Kalman filter algorithm can estimate the constant drift of the gyroscope more accurately. In addition, the Kalman filter algorithm can simultaneously filter the angular acceleration signal and estimate the constant drift of the accelerometer.

The power spectrum is estimated for the angular velocity signals before and after filtering. The result is shown in Figure 10.

As showed from the Figure 10, from the perspective of the frequency domain, the noise after forward linear filtering is 15 dB lower than the power spectrum of the original noise. The noise after the Kalman filtering is reduced by up to 25 dB compared to the power spectrum of the original noise. Therefore, the Kalman filter algorithm can better reduce the gyroscope noise.

#### 3.3.2. Analysis of Dynamic Filtering

Set the sine angular speed command signal with amplitude 2 deg/s and frequency 1 Hz. The filtered result is shown in Figure 11 and Figure 12.

It can be seen from Figure 11 and Figure 12 that both the Kalman filter algorithm and the forward linear filter algorithm can reduce the noise level in the case of dynamic commands. Among them, the Kalman filter algorithm reduces the variance of noise from 1 ${\left(\deg/s\right)}^{2}$ to 0.02 ${\left(\deg/s\right)}^{2}$. The forward linear filtering algorithm reduces the variance of the noise from 1 ${\left(\deg/s\right)}^{2}$ to 0.093 ${\left(\deg/s\right)}^{2}$. Therefore, the filtering effect of the improved Kalman algorithm is significantly better than that of the forward linear algorithm. Moreover, the convergence process of the Kalman filter algorithm in the simulation is also faster.

## 4. Comparison and Analysis of Experimental Results

The experimental test system consists of a computer, a dSpace semiphysical simulation system, and a stable platform. The schematic diagram of the experimental system is shown in Figure 13. The DA interface outputs control commands, which drive the platform motor through the drive. The AD interface receives the accelerometer signal and the gyroscope signal, which is used for filtering and constructing feedback for stable control. The stable platform device is shown in Figure 14. It is a two-axis and two-frame structure which is equipped with sensors such as a MEMS gyroscope and linear accelerometers. The MEMS gyroscope is used to measure platform angular velocity information. Linear accelerometers are used to measure platform acceleration information. Linear accelerometers use symmetrical placement to eliminate the effects of gravitational acceleration.

After that the equipment is installed and debugged, in order to achieve stable control of the stable platform system, the gyroscope signal must be introduced into the feedback channel to form a feedback closed-loop system. Then set different command signals according to different experiments and send them to the driver and motor through dSpace. The gyroscope and line accelerometer are used to acquire the motion signals of the stable platform to observe the corresponding experimental results.

### 4.1. Filtering Experiment

#### 4.1.1. Static Filtering Experiment

Set the angular velocity command signal to zero. The gyroscope signal of the pitch axis of the platform, the acceleration signal, and the filtered signal of the Kalman filter algorithm and the forward linear filter algorithm are collected. The signals are as shown in Figure 15, Figure 16, Figure 17, Figure 18 and Figure 19.

It can be seen from Figure 15, Figure 16, Figure 17, Figure 18 and Figure 19 that both the Kalman filter algorithm and the forward linear filter algorithm can reduce the gyroscope noise level. The Kalman filter algorithm reduces the variance of the noise from 0.323 ${\left(\deg/s\right)}^{2}$ to 0.061 ${\left(\deg/s\right)}^{2}$. The forward linear filtering algorithm reduces the variance of the noise from 0.323 ${\left(\deg/s\right)}^{2}$ to 0.127 ${\left(\deg/s\right)}^{2}$. Moreover, the improved Kalman filter algorithm can estimate the constant drift of the gyroscope more accurately. From the collected gyroscope data, the constant drift of the gyroscope is about 0.45 deg/s, and the drift estimated by the filtering algorithm fluctuates around 0.45 deg/s with an average error of 0.02 deg/s.

In addition, the Kalman filter algorithm can also filter the angular acceleration signal, which reduces the variance of the acceleration noise from 176 ${\left(\deg/s^{2}\right)}^{2}$ to 36 ${\left(\deg/s^{2}\right)}^{2}$. At the same time, it can estimate the constant value drift of the accelerometer. The average error between the estimated value and the actual value is 0.1 $\deg/s^{2}$.

Therefore, from the perspective of time domain signals, the filtering effect of the Kalman algorithm is better than that of the forward linear algorithm.

The power spectrum is estimated for the angular velocity signals before and after filtering. The result is shown in Figure 20.

It can be seen from Figure 20 that in the frequency domain, both filtering algorithms can reduce the power of signal noise, and as the frequency increases, the reduction amplitude is also larger. The noise filtered by the forward linear algorithm is up to 20 dB lower than the power spectrum of the original noise. The filtered noise of the Kalman algorithm is up to 30 dB lower than the power spectrum of the original noise. Therefore, the Kalman filter algorithm has better noise reduction capability from the frequency domain.

#### 4.1.2. Dynamic Filtering Experiment

Set the angular velocity command to a sinusoidal signal with an amplitude of 2 deg/s and a frequency of 1 Hz. The gyroscope signal of the platform’s pitch axis and the filtered signal are shown in Figure 21.

As can be seen from Figure 21, both the Kalman filter algorithm and the forward linear filter algorithm can reduce the noise level in the case of dynamic commands. Among them, the Kalman filter algorithm reduces the variance of noise from 0.56 ${\left(\deg/s^{2}\right)}^{2}$ to 0.11 ${\left(\deg/s^{2}\right)}^{2}$. The forward linear filter algorithm reduces the variance of the noise from 0.56 ${\left(\deg/s^{2}\right)}^{2}$ to 0.31 ${\left(\deg/s^{2}\right)}^{2}$. Therefore, the improved Kalman filter algorithm can better reduce the noise of the gyroscope.

The movements in the above experiments are more conventional movements. In order to fully test the performance of the filter, the platform is artificially rotated randomly without the command signal control; the signal of the gyroscope and the filtered signal are shown in Figure 22.

It can be seen from Figure 22 that in the case of random motion of the platform, the two filtering algorithms still have better filtering effects. It is also apparent from the two filter curves in the figure that the Kalman filter algorithm is superior to the forward linear filter algorithm.

To explore the response speed of the filtering algorithm to the signal in the case of random motion of the platform, the Kalman filter algorithm, the forward linear algorithm, and the commonly used low-pass filtering algorithm are compared. The low-pass filtering algorithm uses a second-order low-pass filter with a bandwidth of 20 Hz. The comparison results are shown in Figure 23.

As can be seen from Figure 23, the filtering effect of the low-pass filter with a bandwidth of 20 Hz is similar to that of the Kalman filter. However, the response speed of the Kalman filter algorithm and the forward linear filter algorithm is 0.01s faster than the low-pass filter with a bandwidth of 20Hz. Therefore, the Kalman filter algorithm and the forward linear filter algorithm have better response speeds to signals than the low-pass filters under the same filtering effect.

### 4.2. Influence of Signal Filtering on System Control Performance

In order to test the improvement of the system control performance by the filtering algorithm, the following experiments were performed on the system using the Kalman filter algorithm and the system without the filter algorithm.

For the system with a filter, the classical controller parameters for speed closed loop are $K_{P}=0.15{\;\mathrm{and}\;K}_{i}=1.5$. This controller is also applied to the unfiltered system. For the above two systems, the zero-angle speed command signal is given; the experimental results are shown in Figure 24, Figure 25 and Figure 26.

As shown in Figure 24, Figure 25 and Figure 26, in the case of using the same controller, there is no significant difference in the angular velocity of the system with or without the filter. This is because the gyroscope is less sensitive to the motion of the platform than the linear accelerometer. Therefore, it can be seen from the line accelerometer signal that the acceleration output noise of the filtered system is significantly smaller than the acceleration output noise of the unfiltered system which can also be seen from the power spectrum of the angular acceleration signal. The power of the acceleration noise of the filtered system is less than the unfiltered power spectrum; the difference is at most 2.5 dB at 250 Hz. During the experiment, under the same controller and command signal, the unfiltered system will have slight jitter, which is the cause of the above phenomenon.

The above phenomenon indicates that the controller parameters applied to the unfiltered system are not suitable. Therefore, the controller parameters need to be re-adjusted for different systems. The more reasonable parameters are: the controller parameters of the filtered system are $K_{P}=0.15{\;\mathrm{and}\;K}_{i}=1.5$, and the controller parameters of the unfiltered system are $K_{P}=0.05{\;\mathrm{and}\;K}_{i}=1$. For the above two systems, a square wave angular velocity command signal with a amplitude of 5 deg/s and a frequency of 1 Hz is given. The experimental results are shown in Figure 27.

As can be seen from Figure 27, for the step signal, both systems can better track the command signal; the systems with filters have a faster step response speed than systems without filters.

Since the stable platform is often affected by disturbances such as bumps, sway, and jitter of the carrier during work, it is necessary to improve the ability of the stable platform to suppress the disturbance. To compare the ability of the stable platform to suppress disturbances before and after filtering, set the zero angular speed command signal, and apply an angular velocity disturbance of ω=2sin(2πt) to the stable platform. The angular error of the platform obtained is shown in Figure 28.

As can be seen from Figure 28, in the case of external disturbance, the variance of the angular error of the system without filter is 0.004 $\deg^{2}$ and the variance of the angular error of the system with the filter is 0.002 $\deg^{2}$. Therefore, the system with a filter has a stronger ability to suppress disturbances.

In summary, the controller parameters are optimized due to the filtering and compensation of the sensor signal by the Kalman filter algorithm. Thereby the control performance of the system and the system’s ability to suppress disturbances are improved.

## 5. Conclusions

In this paper, based on the large noise and drift of MEMS gyroscopes in engineering applications, a Kalman filter algorithm based on information fusion is proposed. It is compared with the forward linear filter algorithm in simulation and experiment. By comparison, it shows the following.
Both filtering algorithms can reduce the noise level of the MEMS gyroscope. However, the filtering effect of the Kalman filter algorithm based on information fusion is better than that of the forward linear filter algorithm, and its noise reduction capability can reach up to 30dB. And it can estimate the constant drift of the MEMS gyroscope more accurately. The average error between the estimated and actual values is 0.02 deg/s.The Kalman filter algorithm based on information fusion can simultaneously reduce the noise level of the accelerometer signal. It can also estimate the constant value drift of the accelerometer, and the average error between the estimated value and the actual value is 0.1 $\deg/s^{2}$.

The processing of the gyroscope signal by the Kalman filtering algorithm based on information fusion not only improves the detection accuracy of the gyroscope, but also optimizes the parameters of the system controller. It also compensates for the drift of the gyroscope, which improves the system’s slow rotation under stable control. The stability of the photoelectric stability platform and the ability to suppress disturbances are improved as a whole, which has important practical significance in practical engineering applications.

## Figures and Tables

**Figure 1 sensors-19-03552-f001:**
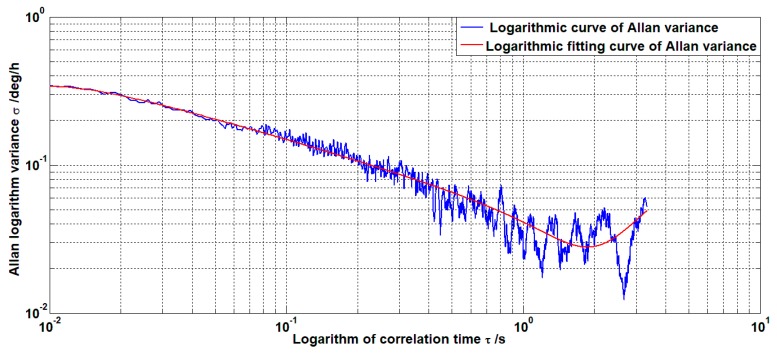
Allan variance logarithmic curve.

**Figure 2 sensors-19-03552-f002:**
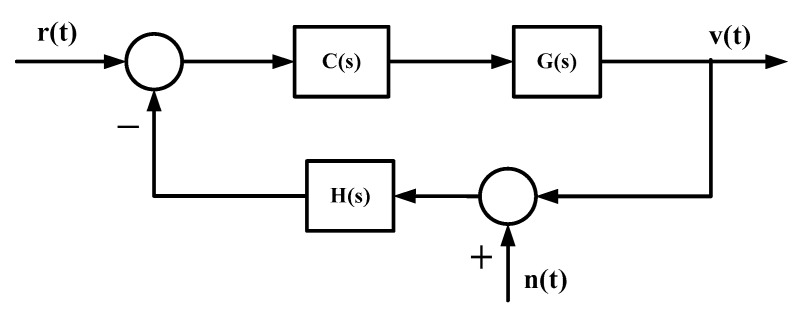
Schematic diagram of the control system structure.

**Figure 3 sensors-19-03552-f003:**
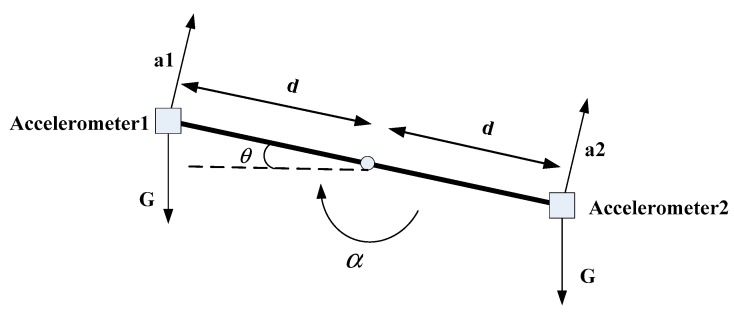
Schematic diagram of the installation of the linear accelerometer.

**Figure 4 sensors-19-03552-f004:**
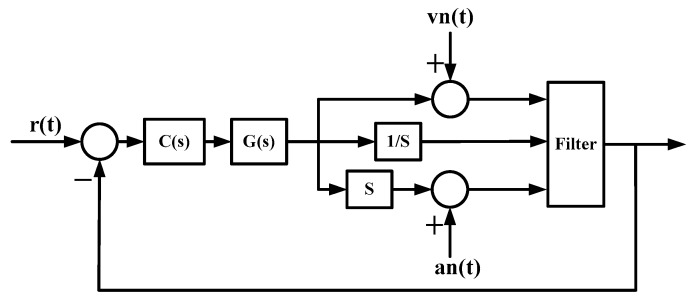
Simulation structure of the filtering algorithm.

**Figure 5 sensors-19-03552-f005:**
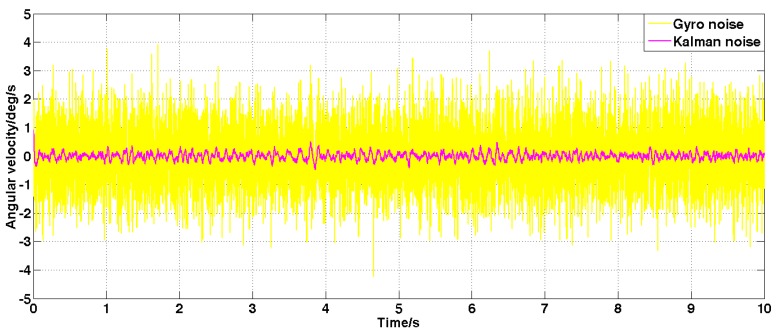
The angular velocity signal after filtering by the Kalman algorithm.

**Figure 6 sensors-19-03552-f006:**
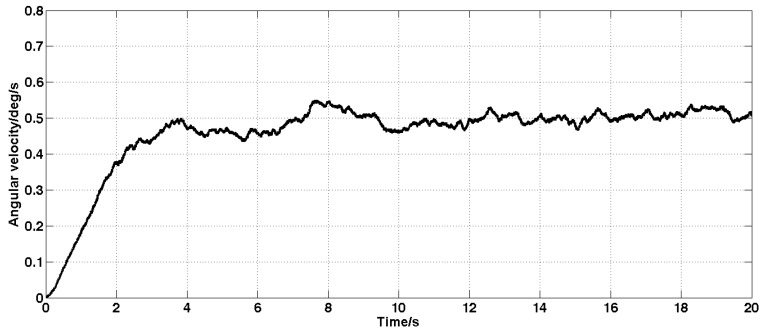
Estimation of gyroscope drift.

**Figure 7 sensors-19-03552-f007:**
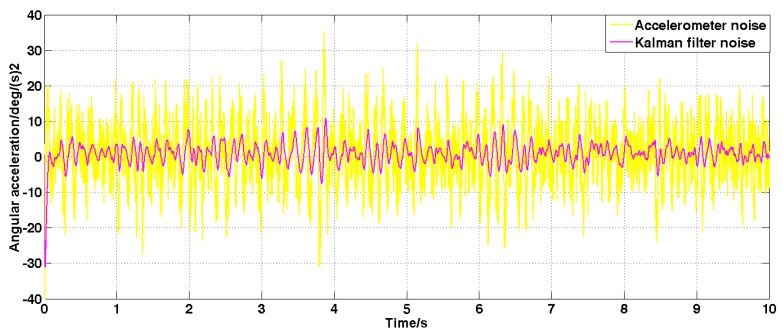
The angular acceleration signal filtered by the Kalman algorithm.

**Figure 8 sensors-19-03552-f008:**
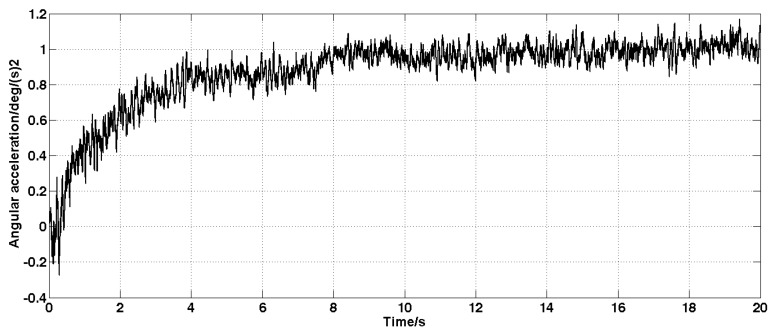
Estimation of accelerometer drift.

**Figure 9 sensors-19-03552-f009:**
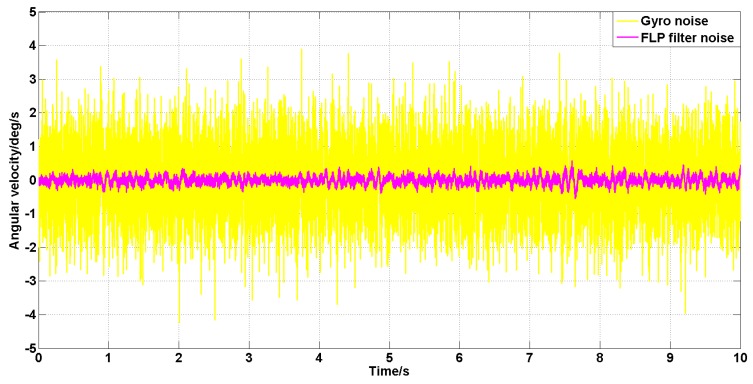
The angular velocity signal filtered by the forward linear algorithm.

**Figure 10 sensors-19-03552-f010:**
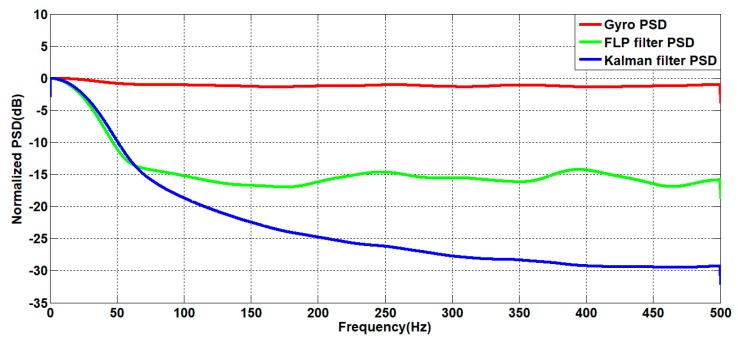
Power spectrum of the angular velocity signal before and after filtering.

**Figure 11 sensors-19-03552-f011:**
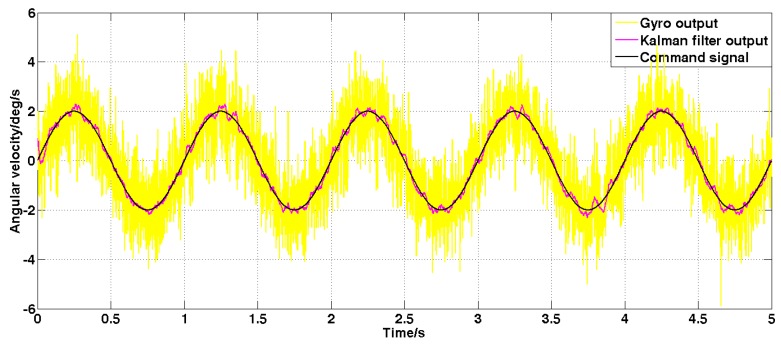
The angular velocity signal filtered by the Kalman algorithm under dynamic commands.

**Figure 12 sensors-19-03552-f012:**
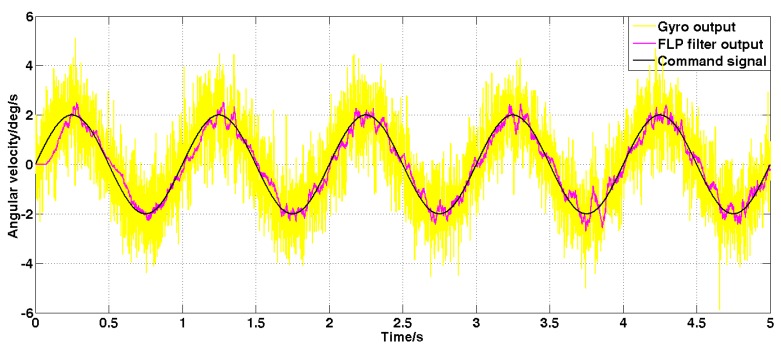
The angular velocity signal filtered by the forward linear algorithm under dynamic command.

**Figure 13 sensors-19-03552-f013:**
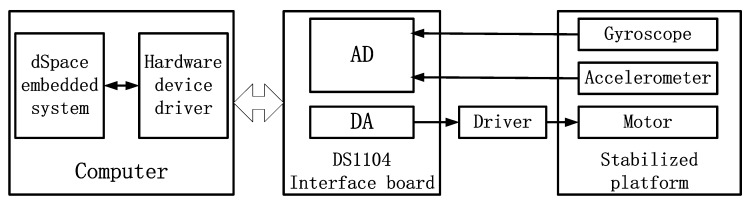
Schematic diagram of the experimental system.

**Figure 14 sensors-19-03552-f014:**
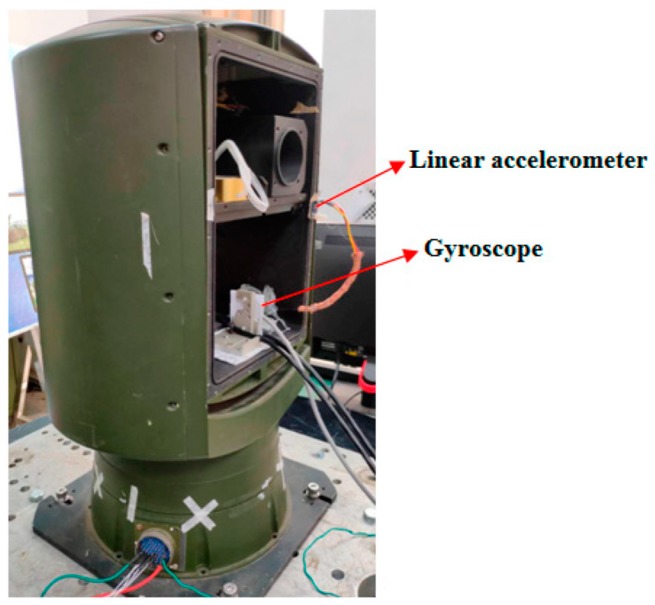
Diagram of the stable platform.

**Figure 15 sensors-19-03552-f015:**
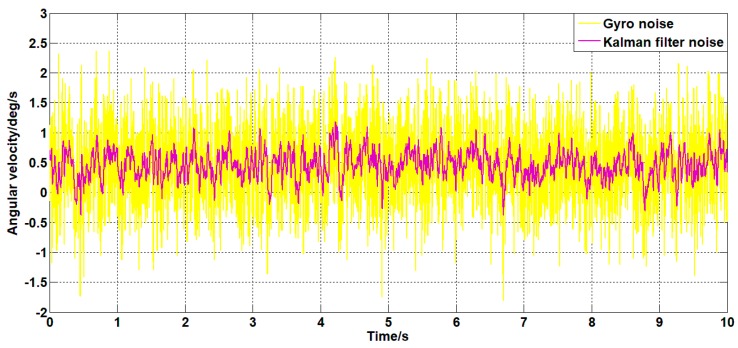
The angular velocity curve after filtering by the Kalman algorithm.

**Figure 16 sensors-19-03552-f016:**
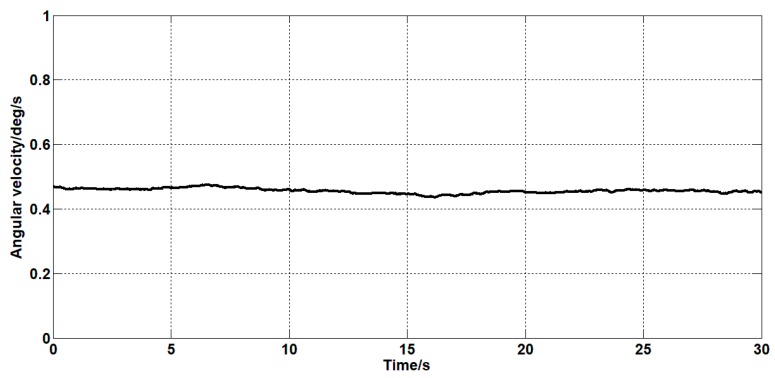
The estimated curve of the drift of the gyroscope.

**Figure 17 sensors-19-03552-f017:**
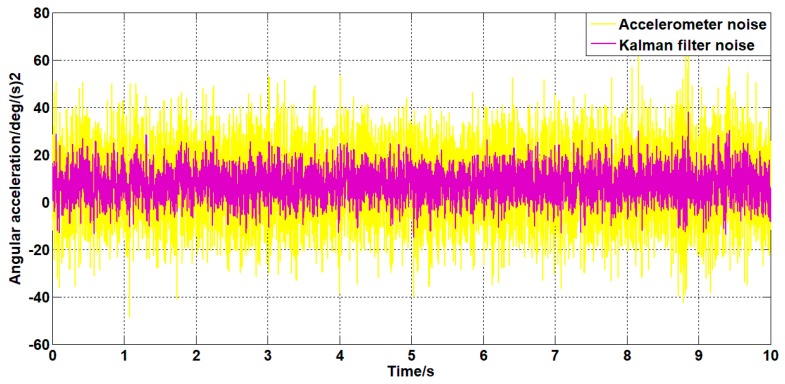
The angular acceleration curve after filtering by the Kalman algorithm.

**Figure 18 sensors-19-03552-f018:**
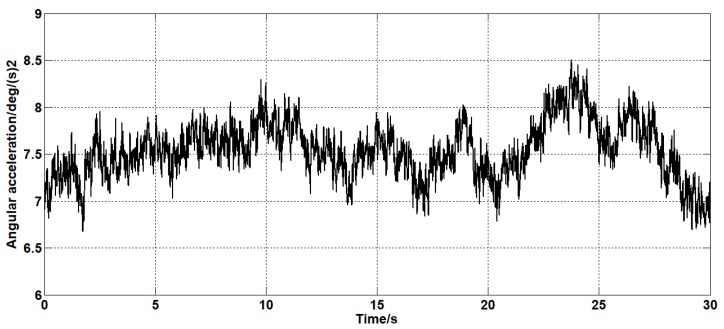
The estimation curve of the drift of the accelerometer.

**Figure 19 sensors-19-03552-f019:**
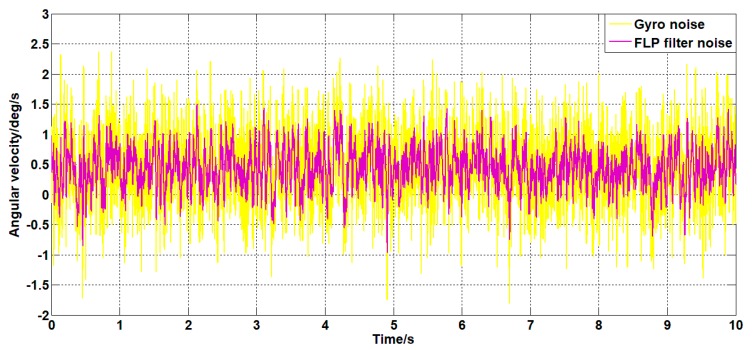
The angular velocity curve filtered by a forward linear algorithm.

**Figure 20 sensors-19-03552-f020:**
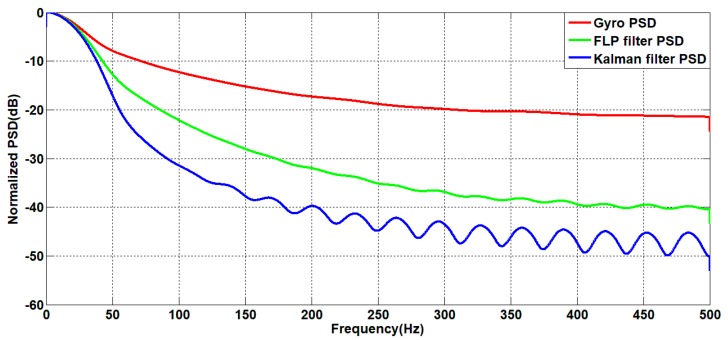
Power spectrum curve before and after filtering.

**Figure 21 sensors-19-03552-f021:**
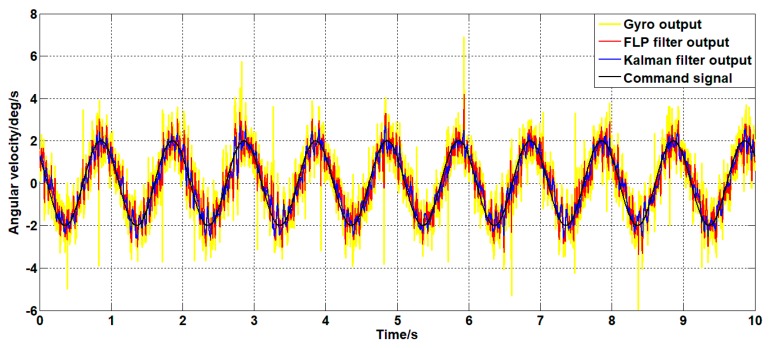
Filter curves for different filtering algorithms under dynamic commands.

**Figure 22 sensors-19-03552-f022:**
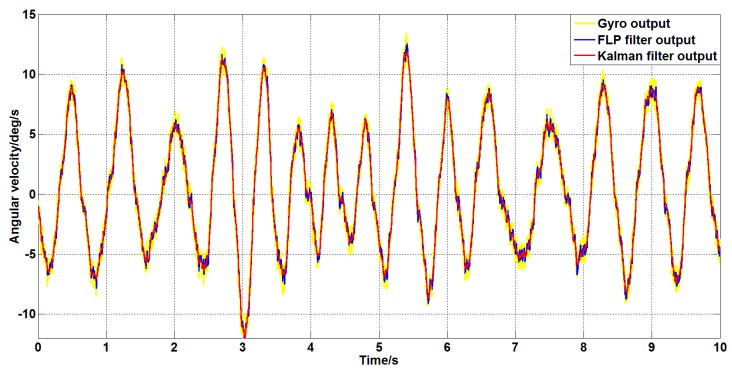
Filtering effect of different filtering algorithms under random motion.

**Figure 23 sensors-19-03552-f023:**
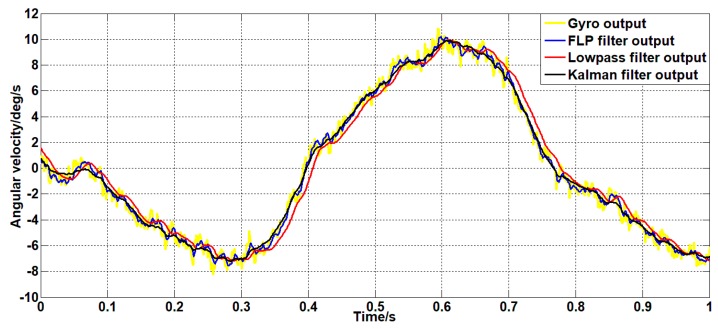
Comparison of response speeds of different filtering algorithms.

**Figure 24 sensors-19-03552-f024:**
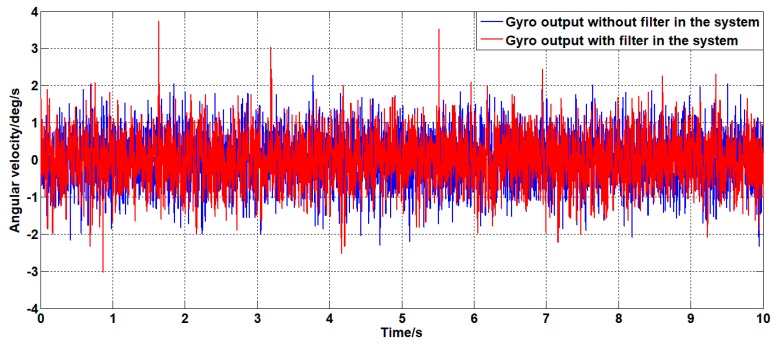
The angular velocity curve of the system output with or without a filter.

**Figure 25 sensors-19-03552-f025:**
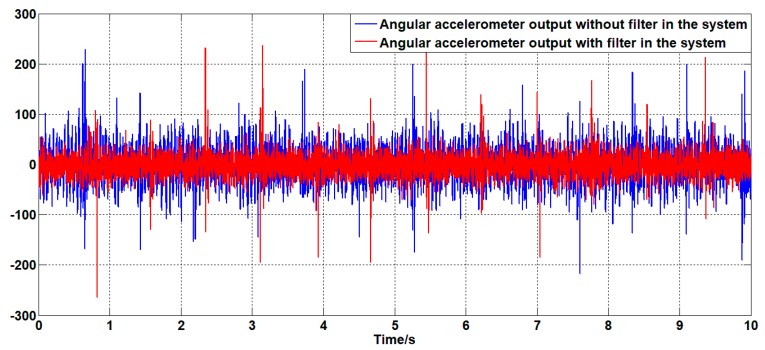
The angular acceleration curve of the system output with or without a filter.

**Figure 26 sensors-19-03552-f026:**
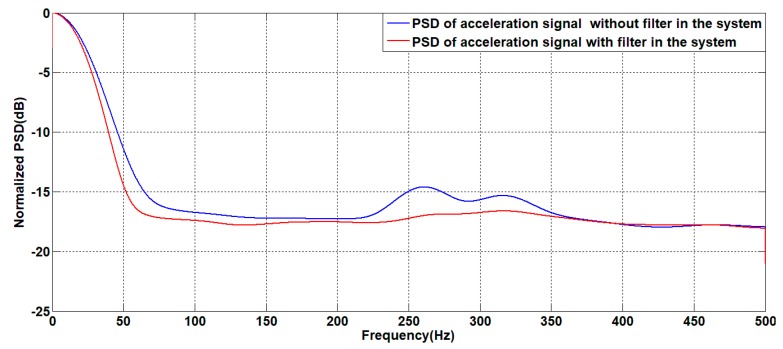
Power spectrum of the angular acceleration of the system output with or without a filter.

**Figure 27 sensors-19-03552-f027:**
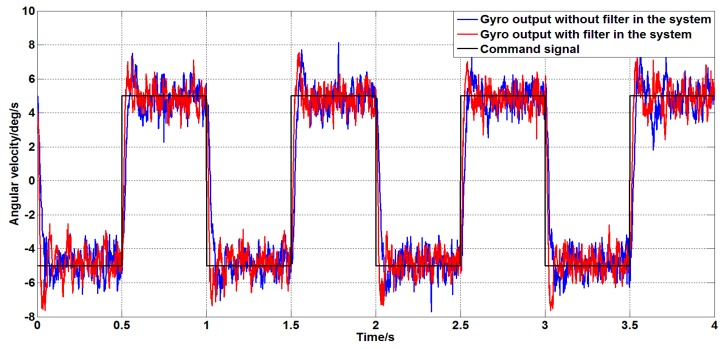
The step response of a system with or without a filter.

**Figure 28 sensors-19-03552-f028:**
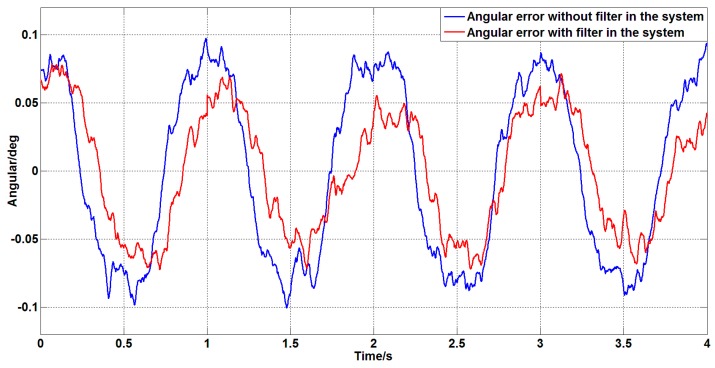
The angular error of the system with or without a filter under disturbance.

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
