# Peer review of "Research on Filtering Algorithm of MEMS Gyroscope Based on Information Fusion"

_sensors, 2019, doi:10.3390/s19163552_

Round 1
Reviewer 1 Report
This ideas in the paper are innovative and the theoretic results obtained have many potentials in applications. By comparing, it is proved that the proposed method is more effective in reducing noise and drift, which makes the paper more convincing. The chapter design is reasonable and the presentation is expressive in whole although there are some questions which make the paper not relatively rigorous. Ten suggestions are listed as follows.
(1) In page 6 line 194, the proof of controllable and observable of the system is too insufficient and hurried because of the absence of B matrix.
(2) In page 9 line 239, the contradistinction of the data from two filters is conflicted with your conclusion that the improved Kalman filter’s ability of reducing noise is better than Forward Linear Filter, so you’d better check it was due to writing wrong data mistakenly or making a wrong conclusion.
(3) In page 9 line 243, the absence of correlation diagrams of Forward Linear Filter with respect to gyro drift filtering makes your contradistinctions weak, so they should be put in this paper.
(4) In page 11, the whole chapter of Filtering Experiment is almost repeated with before simulation and there’s no need for the same content to take up a large amount of space or you have to change a more suitable experiment.
(5) The new method which you proposed is not suitable and appropriate for its name “improved Kalman filter”, because you do use Kalman filter instead of improving Kalman filter. So, you’d better change a more suitable name or design some experiment of Kalman filter used in other papers to make a contradistinction.
(6) You mainly emphasis on the introduction of theory, but a quality paper asks for adequate experiment contents which you need to enrich, for example, experimental operation and so on.
(7) You need to notice parallel problem of serial number if they are not concatenation relationship. Just like, in page 6, the number (2) (3) is the explanation of (1) rather than concatenation. So, you need to adjust this part of the structure.
(8) The citation of the literature does not highlight the urgent problem you need to solve
(9) In page 16, the head of figure 25 has a spelling mistake.
(10) In page 17, the conclusion of the article is too superficial, and it should be the author’s deeper insight into the problem from the whole, not just a summary of the experimental results.
Reviewer 2 Report
The authors introduce an improved Kalman filter to reduce the noise and compensate the constant drift of gyroscopes. The innovation of this paper is to improve the model of Kalman filter, and the constant drift and the information of accelerometer are augmented into the state vector.
During the reading, I have some specific concerns:
1. For the introduced model of Kalman filter, I miss the mathematical derivation, which indicates, that the reasonableness of this improvement are remained unclear and indeterminate.
2. The paper has some logical mistakes.
Line 236~ Line 244: the improved Kalman filter algorithm reduces the variance of the noise from 1 (deg/s)2 to 0.0156 (deg/s)2. The forward linear filtering algorithm reduces the variance of the noise from 1 (deg/s)2 to 0.0119 (deg/s)2. But the paper concluded the improved Kalman filter have the better performance. Please comment.
3. The introduction doesn’t have a detailed review on the current researches. The related researches are simply listed and introduced, and the problems of these researches are not analyzed.
4. There are some spelling mistakes. Line 115, the “figure 2” should be revised as “Figure 2”; in the Figure 8, the “nooise” should be revised as “noise”. The authors must check all of the manuscript carefully and correct them.
5. Some mathematical expressions are not clear and proper, (eg. Equation (19)). The vectors and matrices are supposed to be Italic and Bold. The authors must check all of the manuscript carefully and correct them.
6. In the Fig. 21 and Fig. 22, the red lines represent the filtered signals. But the figures show that, in the filtered signals, the outliers appear at the sampling points where the outliers do not exist in the original signals. Please clearly indicate this problem.
In general, the overall contribution remains scientifically weak and technically questionable. The authors should supplement the content further. Besides, the issues listed above have to be explained and revised.
Reviewer 3 Report
1. The line accelerometer can get angle information only when the object is static. I suggest that the author give a detailed introduction on how to get the angle information when the object is moving.
2. The theory in Section 2 “Analysis of MEMS Gyroscope Error” is classic, I suggest the author should simplify this part.
3. I suggest the author should add some complex motion experiments (The angular velocity command signal in this paper is all regular). Also the author only compare his/her proposed algorithm with FLP. I suggest the author should compare this algorithm with other algorithms too if possible.
Reviewer 4 Report
Page 7, line 216, H(s) is never mentioned in the texts nor figures. I guess figure 3 should contain H(s) anywhere.
For dynamic command input, is response speed with Kalman filter and FLP filter sufficient?
How frequent input does the gyro with two filter respond to?
Why there is no significant difference between filtered and unfiltered system in the angular velocity? (About Figure 21 and 22) More explanation will be needed.
Page16, line 369, "angular velocity disturbance" is not so clear. Which kinds of disturbance is expected in this case?
Round 2
Reviewer 2 Report
The authors have sufficiently and satisfactorily improved the quality of the paper with respect to its previous submission. Thus, the paper should be accepted as it is.